

# Cytokine profiles of mild-to-moderate SARS-CoV-2 infected and recovered pre-vaccinated individuals residing in Indonesia

Ni Luh Ayu Megasari[1,2], Siti Qamariyah Khairunisa[1], Radita Yuniar Arizandy[1], I. Komang Evan Wijaksana[3] and Citrawati Dyah Kencono Wungu[1,4]

[1] Institute of Tropical Disease, Airlangga University, Surabaya, Indonesia
[2] Postgraduate School, Airlangga University, Surabaya, Indonesia
[3] Department of Periodontology, Faculty of Dental Medicine, Airlangga University, Surabaya, Indonesia
[4] Department of Physiology and Medical Biochemistry, Faculty of Medicine, Airlangga University, Surabaya, Indonesia

Corresponding author
Citrawati Dyah Kencono Wungu, citrawati.dyah@fk.unair.ac.id

## ABSTRACT

**Background**. Accumulating evidence suggests the involvement of cytokine-mediated inflammation, in clinical severity and death related to SARS-CoV-2 infection, especially among pre-vaccinated individuals. An increased risk of death was also described among SARS-CoV-2 recovered individuals, which might be correlated with prolonged inflammatory responses. Despite being among the countries with the highest cumulative deaths due to COVID-19, evidence regarding cytokine profiles among SARS-CoV-2 infected and recovered pre-vaccinated individuals in Indonesia is scarce. Thus, this study aimed to describe the cytokines profiles of pre-vaccinated individuals residing in Indonesia, with mild-to-moderate SARS-CoV-2 infection and those who recovered.

**Methods**. Sixty-one sera from 24 hospitalized patients with mild-to-moderate SARS-CoV-2 infection, 24 individuals recovered from asymptomatic-to-moderate SARS-CoV-2 infection, and 13 healthy controls unexposed to SARS-CoV-2 were used in this study. Quantification of serum cytokine levels, including IL-6, IL-8, IP-10, TNF-$\alpha$, CCL-2, CCL-3, CCL-4, and CXCL-13, was performed using a Luminex multi-analyte-profiling (xMAP)-based assay.

**Results**. The levels of IL-8 along with CCL-2 and CCL-4, were significantly higher ($p \leq 0.01$) in hospitalized patients with mild-to-moderate SARS-CoV-2 infection and recovered individuals compared to healthy controls. However, no significant difference was observed in these cytokine levels between infected and recovered individuals. On the other hand, there were no significant differences in several other cytokine levels, including IL-6, IL-10, TNF-$\alpha$, CCL-3, and CXCL-13, among all groups.

**Conclusion**. IL-8, CCL-2, and CCL-4 were significantly elevated in pre-vaccinated Indonesian individuals with mild-to-moderate SARS-CoV-2 infection and those who recovered. The cytokine profiles described in this study might indicate inflammatory responses not only among SARS-CoV-2 infected, but also recovered individuals.

## INTRODUCTION

Coronaviruses were considered a public health concern following the outbreak of severe acute respiratory syndrome coronavirus (SARS-CoV) and Middle East Respiratory Syndrome Coronavirus (MERS-CoV) in 2002 and 2012, respectively (*Cui, Li & Shi, 2019*; *Hu et al., 2021*). At the end of 2019, a novel coronavirus started to cause an outbreak of unusual viral pneumonia emerged in Wuhan, China. It was then designated as SARS-CoV-2, the etiological agent of coronavirus disease 2019 (COVID-19). Following the rapid transmission all over the world, the World Health Organization (WHO) declared the SARS-CoV-2 outbreak as a Public Health Emergency of International Concern (*Cucinotta & Vanelli, 2020*; *Hu et al., 2021*).

Up to February 8, 2023, 754,816,715 confirmed COVID-19 cases and 6,830,232 deaths worldwide were reported by the WHO. Indonesia was among the ten countries with the highest cumulative deaths, with 160,847 deaths reported (*World Health Organization, 2023*). As an effort to mitigate the pandemic, the Indonesian authority kickstarted the national vaccination program on January 13, 2021 (*Soegiarto et al., 2023*). Before that, the seroprevalence of SARS-CoV-2 infection was reported at around 11.4% in East Java (*Megasari et al., 2021*), and the WHO reported 818,386 cumulative COVID-19 cases or 299.2 cases/100,000 population, with 23,947 cumulative deaths or 8.8 deaths/100,000 population in Indonesia (*World Health Organization, 2021a*). Administration of vaccine is suggested to prevent the severity of disease and reduce COVID-19-related deaths (*Rahmani et al., 2022*; *Huang & Kuan, 2022*).

Accumulating evidence suggested the involvement of cytokine-mediated inflammation, mainly described as cytokine storms, in COVID-19 severity and death, especially among pre-vaccinated individuals. Cytokine storm is an acute overproduction and release of pro-inflammatory cytokines both locally and systemically, and may trigger acute respiratory distress syndrome (ARDS), a major cause of death among COVID-19 patients (*Mehta et al., 2020*; *Hu et al., 2022*; *Jiang et al., 2022*; *Montazersaheb et al., 2022*). Several cytokines overly expressed or elevated in SARS-CoV-2 infected individuals including interleukin (IL) 1 (IL-1), IL-1 beta ($\beta$), IL-4, IL-6, IL-7, IL-8, IL-10, IL-15, IL-17, tumor-necrosis-factor (TNF) alpha (TNF-$\alpha$), C-C motif chemokine ligand (CCL) 2 (CCL-2), CCL-3, CCL-4, CCL-7, CCL-20, C-X-C motif chemokine ligand (CXCL) 1 (CXCL-1), CXCL-3, CXCL-10, and CXCL-13 (*Liao et al., 2020*; *Zhou et al., 2020*; *Chua et al., 2020*; *Xiong et al., 2020*; *Coperchini et al., 2021*; *Hu et al., 2022*; *Frisoni et al., 2022*; *Montazersaheb et al., 2022*).

Most studies described cytokine profiles mainly in severe, critical, and deceased pre-vaccinated COVID-19 patients (*Mulchandani, Lyngdoh & Kakkar, 2021*; *Hu et al., 2022*; *Frisoni et al., 2022*). However, approximately 80% of infected individuals who became symptomatic develop only mild or moderate illness, even before vaccination (*Blair et al., 2021*; *World Health Organization, 2021b*). Thus, cytokine profiles among pre-vaccinated mild-to-moderate SARS-CoV-2 infected individuals also need to be elucidated.

A study in Estonia, mainly among pre-vaccinated individuals, identified an increased risk of death among people with a history of SARS-CoV-2 infection, with more than three times the risk of dying over the following year (*Uusküla et al., 2022*), which might be due

to prolonged inflammation despite the recovery. As the COVID-19 pandemic situation continues, cytokine profiles among pre-vaccinated, recovered individuals might need to be studied.

Despite being among the countries with the highest cumulative deaths due to COVID-19, evidence regarding cytokine profiles among mild-to-moderate SARS-CoV-2 infected pre-vaccinated individuals is very limited (*Ramatillah et al., 2022*), and no available study in recovered pre-vaccinated individuals in Indonesia to date. Up to February 2024, the Ministry of Health of Indonesia reported less than 90% and 80% of Indonesian individuals received first and second-dose of COVID-19 vaccination, respectively. Therefore, the result of this study might be beneficial for pre-vaccinated individuals in Indonesia (*Indonesian Ministry of Health, 2024*). This study aimed to describe cytokines profiles, especially IL-6, IL-8, IP-10, TNF-$\alpha$, CCL-2, CCL-3, CCL-4, and CXCL-13, of pre-vaccinated individuals residing in Indonesia, with mild-to-moderate SARS-CoV-2 infection and those who recovered.

## MATERIALS & METHODS

### Ethics statement and sample collection

This study was ethically approved by the Ethical Clearance Commission of Health Research, Faculty of Dental Medicine, Universitas Airlangga (Ethical clearance certificate no. 735/HRECC.FODM/2022). Frozen serum samples collected from hospitalized patients with mild-to-moderate SARS-CoV-2 infection, individuals recovered from asymptomatic-to-moderate SARS-CoV-2 infection, and healthy controls unexposed of SARS-CoV-2 (collected prior to the COVID-19 pandemic) were used in this study.

Samples from infected and recovered individuals were collected from June to December 2020 at a private hospital and a university research center in East Java, Indonesia, respectively. SARS-CoV-2 infection in hospitalized patients was determined by positive detection of SARS-CoV-2 RNA using nucleic acid amplification test (NAAT). Samples from infected individuals were collected within three days after symptom onset. Individuals with positive anti-SARS-CoV-2 immunoglobulin G (Ig-G), negative SARS-CoV-2 NAAT, and exhibiting no COVID-19-related symptoms for a minimum of two-months prior to sample collection, were considered recovered from SARS-CoV-2 infection. SARS-CoV-2 infection was classified according to the criteria established by The National Institutes of Health (*COVID-19 Treatment Guidelines Panel, 2022*). Samples from healthy controls were collected in early 2019. All participants received no COVID-19 vaccine at the time of sample collection. Written informed consent was obtained from all participants prior to sample collection. Following sample collection, serums were prepared and stored as previously described (*Megasari et al., 2021*). Demographic and clinical characteristics of study participants were retrieved from medical records and administered questionnaires.

### Serum cytokine levels quantification

Quantification of serum cytokine levels, including IL-6, IL-8, IP-10, TNF-$\alpha$, CCL-2 (MCP-1), CCL-3 (MIP-1$\alpha$), CCL-4 (MIP-1$\beta$), and CXCL-13 (BCA-1), was performed using the MILLIPLEX Human Cytokine kit (Sigma-Aldrich, Singapore), a Luminex
multi-analyte-profiling (xMAP)-based assay. All procedures were carried out following the manufacturer's instructions. Data were acquired on a validated and calibrated Luminex MAGPIX system (Luminex Corporation, Hayward, CA, USA) and analyzed with xPONENT software (Luminex Corporation).

## Statistical analysis

Statistical analysis was performed using GraphPad Prism 9 (GraphPad Software, La Jolla, CA, USA). A normality test was performed using Shapiro–Wilk. Based on the results of the normality test, the comparisons of cytokine levels between sex groups were analyzed using the Mann–Whitney U-test, while the correlation analysis between age and cytokine levels was performed using Spearman's test. Results with $p < 0.05$ were considered significant.

## RESULTS

### Demographic characteristics of research participants

Sixty-one samples from 24 hospitalized patients with mild-to-moderate SARS-CoV-2 infection, 24 individuals recovered from asymptomatic-to-moderate SARS-CoV-2 infection, and 13 healthy controls unexposed of SARS-CoV-2 were used in this study. Most participants (59/61; 96.72%) were of Javanese ethnicity. Mean age was $50.04 \pm 12.88$, $42.38 \pm 13.81$, and $32.46 \pm 7.68$ for infected, recovered, and control groups, respectively. Twenty-three (37.7%) participants were male and 38 (62.3%) were female. Demographic characteristics of the participants are presented in Table 1.

### Serum cytokine levels

The levels of IL-8 (Fig. 1) along with CCL-2 and CCL-4 (Fig. 2) were significantly higher ($p \leq 0.01$) in hospitalized patients with mild-to-moderate SARS-CoV-2 infection and recovered individuals compared to healthy control. However, no significant difference was observed in these cytokine levels between infected and recovered individuals. On the other hand, there were no significant differences in several other cytokine levels, including IL-6, IL-10, TNF-$\alpha$, CCL-3, and CXCL-13 among all groups. The comparison of the cytokine levels in each group can be seen in Table 2. The levels of IL-10 were significantly different between males and females in the recovered group, while CXCL-13 levels were correlated to the age of individuals in the recovered group. Cytokine levels of different sex and groups are presented in Table 3.

## DISCUSSION

The primary aim of this study was to describe cytokine profiles among mild-to-moderate SARS-CoV-2 infected and recovered pre-vaccinated individuals in Indonesia. Our finding suggests the possibility of prolonged elevation of cytokine levels, especially IL-8, CCL2, and CCL4, among individuals recovered from SARS-CoV-2 infection.

Interleukin-8 was identified as a sensitive biomarker in mild and severe COVID-19 patients and suggested as a better indicator for overall COVID-19 disease status. The levels of IL-8 in the sera of mild COVID-19 patients were significantly higher than in non-infected individuals and further elevated in severe patients (*Li et al., 2020*; *Huang et al., 2020*; *Chen et*

**Table 1  Summary of participants characteristics by study groups.**

| Characteristics | Mild-to-moderate SARS-CoV-2 infection | | Recovered | | Healthy controls | | P-value |
|---|---|---|---|---|---|---|---|
| | n = 24 | % | n = 24 | % | n = 13 | % | |
| Age (mean ± SD) | 50.04 ± 12.88 | | 42.38 ± 13.81 | | 32.46 ± 7.68 | | 0.001[*] |
| Sex | | | | | | | 0.321 |
| Male | 7 | 29.17 | 12 | 50 | 4 | 30.77 | |
| Female | 17 | 70.83 | 12 | 50 | 9 | 69.23 | |
| Ethnicity | | | | | | | 0.043[*] |
| Javanese | 24 | 100 | 24 | 100 | 11 | 84.61 | |
| Madurese | 0 | | 0 | | 2 | 15.39 | |

Notes.

*Significant at $p < 0.05$.

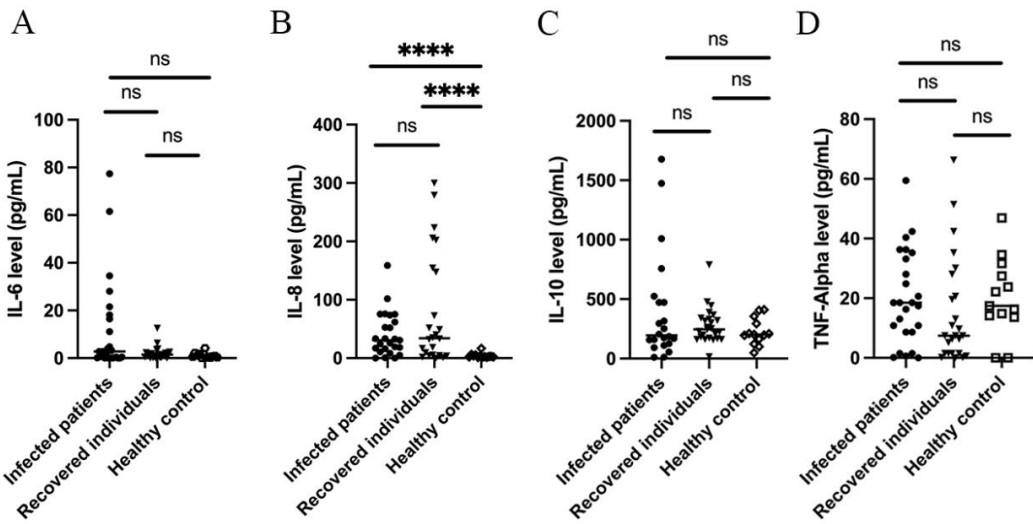

**Figure 1  Cytokine levels of (A) IL-6; (B) IL-8; (C) IL-10; and (D) TNF-α.**

*al., 2020; Kesmez Can et al., 2021*). However, compared to serum IL-6, which is noticeably elevated in severe cases, serum IL-8 was easily detectable in mild infection (*Li et al., 2020; Chen et al., 2020*). Higher IL-8 was also observed among individuals who recovered from COVID-19 up to eight months following mild-to-moderate infection, compared to healthy, unexposed control (*Phetsouphanh et al., 2022*).

Consistent with previous studies, we observed higher IL-8 levels not only among mild-to-moderate SARS-CoV-2 infected Indonesian individuals but also in those who recovered. IL-8, known as neutrophil chemotactic factor, is a pro-inflammatory cytokine that recruits neutrophils to the site of infection and has been associated with tissue damage (*Ma et al., 2021; Phetsouphanh et al., 2022*). Increased expression of IL-8 has been characterized in many chronic inflammatory respiratory conditions, including acute lung injury, chronic obstructive pulmonary disease, and lung epithelial cells injury (*Qazi, Tang & Qazi, 2011*).

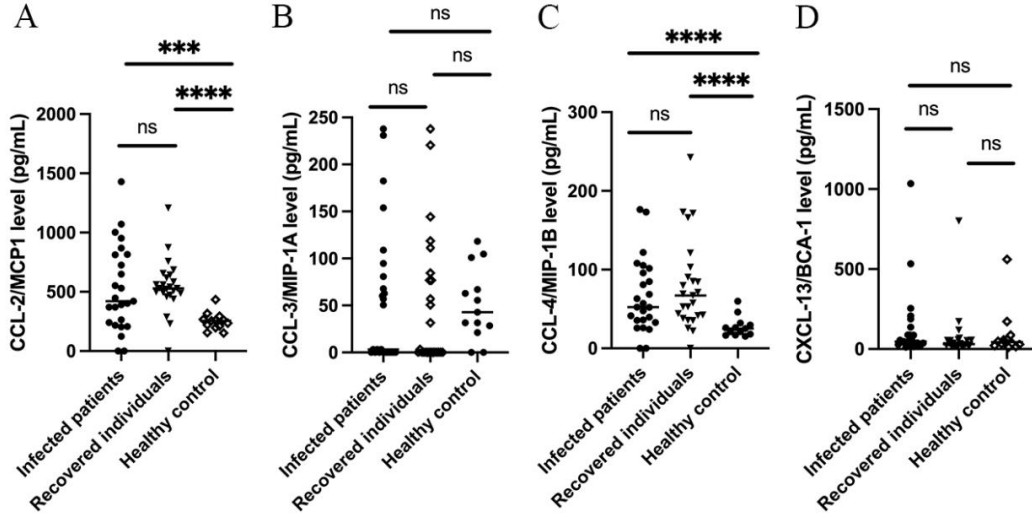

**Figure 2** Cytokine levels of (A) CCL-2/ MCP1; (B) CCL-3/MIP-1α; (C) CCL-4/MIP-1β; and (D) CXCL-13/BCA-1.

**Table 2  Cytokine levels between mild-to-moderate SARS-CoV-2 infection, recovered individuals, and healthy controls.**

| Parameter | Mild-to-moderate SARS-CoV-2 infection | | Recovered | | Healthy controls | | p value |
|---|---|---|---|---|---|---|---|
| | Mean (SD) | Median (IQR) | Mean (SD) | Median (IQR) | Mean (SD) | Median (IQR) | |
| IL-6 (pg/mL) | 11.64 (20.03) | 2.79 (0.10–17.31) | 32.12 (129.7) | 1.53 (0.39–3.055) | 1.12 (1.162) | 0.75 (0.27–1.8) | 0.41 |
| IL-8 (pg/mL) | 40.21 (37.67) | 30.97 (13.53–67.92) | 451.1 (1,864) | 35.26 (5.040–178.3) | 4.21 (4.11) | 2.99 (1.84–4.84) | 0.0005[*] |
| IL-10 (pg/mL) | 4387.4 (13,489) | 253.06 (141.73–882.8) | 275.04 (149.46) | 245.62 (169.71–336.34) | 224.28 (112.73) | 201.87 (143.18–324.89) | 0.59 |
| TNF-α (pg/mL) | 20.06 (15.61) | 18.49 (8.6–32.44) | 14.98 (17.84) | 7.46 (1.4–24.32) | 20.22 (13.13) | 17.41 (13.88–29.55) | 0.22 |
| CCL-2/MCP-1 (pg/mL) | 522.78 (360.95) | 420.55 (229.43–814.44) | 668.17 (640.32) | 529.38 (479.97–649.36) | 251.01 (73.22) | 255.32 (203.48–281.14) | 0.0004[*] |
| CCL-3/MIP-1α (pg/mL) | 55.89 (74.73) | 3.2 (0-87.71) | 48.52 (70.54) | 0 (0-80.67) | 51.14 (38.43) | 42.76 (24.88–83.97) | 0.50 |
| CCL-4/MIP-1β (pg/mL) | 66.04 (46.34) | 52.40 (34.27–98.75) | 78.93 (57.09) | 67.06 (39.85–96.73) | 27.29 (12.86) | 25.15 (17.51–30.95) | 0.0005[*] |
| CXCL-13/BCA-1 (pg/mL) | 130.77 (223.57) | 46.58 (28.39–124.5) | 74.74 (158.67) | 30.86 (20.35–55.05) | 90.35 (147.17) | 43.65 (25.43–73.5) | 0.18 |

**Notes.**

*Significant at $p < 0.05$.

Megasari et al. (2024), *PeerJ*, DOI 10.7717/peerj.17257

**Table 3  Cytokine levels between different sex and age group.**

| Parameter | Infected individuals | | | | Recovered individuals | | | | Healthy control | | | |
|---|---|---|---|---|---|---|---|---|---|---|---|---|
| | Male | Female | *p* value for sex[a] | *P* value for age[b] | Male | Female | *p* value for sex[a] | *P* value for age[b] | Male | Female | *p* value for sex[a] | *P* value for age[b] |
| IL-6 (pg/mL) | 0.1 (28.08) | 2.79 (14.16) | 0.390 | 0.626 | 1.32 (9.93) | 1.79 (2.28) | 0.402 | 0.618 | 1.37 (1.65) | 0.6 (1.18) | 0.355 | 0.628 |
| IL-8 (pg/mL) | 33.58 (64.21) | 24.43 (44.61) | 0.409 | 0.610 | 13.09 (184.64) | 50.44 (159.31) | 0.119 | 0.561 | 3 (2.11) | 2.82 (4.01) | 0.758 | 0.943 |
| IL-10 (pg/mL) | 157.87 (432.31) | 316.56 (1074.08) | 0.172 | 0.377 | 173.87 (131.03) | 317.88 (193.48) | 0.021* | 0.590 | 181.08 (123.36) | 210.05 (221.77) | 0.123 | 0.404 |
| TNF-$\alpha$ (pg/mL) | 16.33 (34.77) | 18.49 (23.38) | 0.633 | 0.674 | 6.89 (16.43) | 9.72 (23.83) | 0.340 | 0.567 | 27.71 (37.15) | 16.31 (10.93) | 0.315 | 0.580 |
| CCL-2/MCP-1 (pg/mL) | 402.85 (605.89) | 446.82 (525.82) | 0.568 | 0.372 | 531.59 (330.41) | 529.11 (165.4) | 0.686 | 0.712 | 245.47 (51.52) | 262.28 (122.6) | 0.537 | 0.553 |
| CCL-3/MIP-1$\alpha$ (pg/mL) | 50.47 (67.89) | 3.2 (101.75) | 0.974 | 0.769 | 1.6 (102.78) | 15.8 (82.6) | 0.926 | 0.896 | 85.85 (82.14) | 31.6 (44.88) | 0.122 | 0.950 |
| CCL-4/MIP-1$\beta$ (pg/mL) | 57.07 (56.41) | 52.4 (72.55) | 0.949 | 0.812 | 59.93 (70.32) | 68.77 (63.71) | 1.000 | 0.058 | 23.14 (10.6) | 26.15 (21.82) | 0.758 | 0.511 |
| CXCL-13/BCA-1 (pg/mL) | 37.62 (58.86) | 55.46 (133.55) | 0.446 | 0.748 | 31.58 (33.56) | 30.86 (43.94) | 0.908 | 0.008* ($\rho = -0.527$) | 58.74 (49.79) | 35.65 (87.13) | 0.440 | 0.149 |

**Notes.**

Descriptive data is in the form of median (IQR).

[a] Mann–Whitney test.

[b] Spearman's correlation test.

*Significant at $p < 0.05$.

$\rho$, correlation coefficient.

CCL-2 or monocyte chemoattractant protein-1 (MCP-1) is among the cytokines involved in the regulation of monocytes or macrophages migration and infiltration into infected tissues, including lungs (*Coperchini et al., 2021*; *Ranjbar et al., 2022*). The levels of CCL-2 were reported to be elevated in COVID-19 patients compared to healthy individuals, especially those infected with ancestral, alpha, and omicron SARS-CoV-2 variants (*Huang et al., 2020*; *Zhao et al., 2020*; *Korobova et al., 2022*). In a longer duration of SARS-CoV-2 infection, from 14 days to four weeks, CCL-2 levels were reported to be similar between severe and mild infection (*Xu et al., 2020*; *Zhao et al., 2020*). De-Oliviera-Pinto et al. reported no significant difference in CCL-2 levels between symptomatic and asymptomatic SARS-CoV-2 infected individuals, and also those who recovered (*De-Oliveira-Pinto et al., 2022*). Similar to the findings, we found that the CCL-2 levels of mild-to-moderately SARS-CoV-2 infected pre-vaccinated individuals were significantly different compared to healthy, unexposed controls, but not with individuals recovered.

CCL-4, or Macrophage inflammatory protein-1 $\beta$(MIP-1 $\beta$), is a chemoattractant for several immune cells, particularly T cells (*Li, Yeung & Schooling, 2021*). Similar to CCL-2, CCL-4 was also reported to be elevated in COVID-19 patients and correlated to several variants (*Huang et al., 2020*; *Zhao et al., 2020*; *Korobova et al., 2022*). In mild, severe, and fatal cases of SARS-CoV-2 infection, the expression of CCL-4 was shown to be upregulated. However, significantly higher expression was observed in mild cases and suggested the association of CCL-4 with recovery and resolution of inflammation through the activation of cytotoxic T cells (*Xu et al., 2020*; *Zhao et al., 2020*). However, CCL-4 levels among COVID-19 recovered individuals have not been described. CCL-4 levels among mild-to-moderate SARS-CoV-2 infected individuals in this study were significantly higher than those of healthy controls, as reported in other studies. We also found that individuals recovered from SARS-CoV-2 infection exhibited higher CCL-4 levels compared to healthy controls.

Despite being one of the most extensively studied and reviewed (*Huang et al., 2020*; *Mojtabavi, Saghazadeh & Rezaei, 2020*; *Coomes & Haghbayan, 2020*; *Chen et al., 2020*), this study found that IL-6 levels among mild-to-moderate COVID-19 patients, recovered individuals, and healthy controls were not significantly different. Several meta-analyses concluded that IL-6 levels are positively correlated with disease severity and adverse clinical outcomes (*Mojtabavi, Saghazadeh & Rezaei, 2020*; *Coomes & Haghbayan, 2020*); however, these studies observed no significant difference between IL-6 levels of mild and severe infection, especially in the early stage of infection (*Xu et al., 2020*; *Zhao et al., 2020*).A study by Phetsouphanh et al., which observed patients recovered up to 4 months from COVID-19, also found no significant difference in IL-6 levels between recovered individuals and healthy controls (*Phetsouphanh et al., 2022*). The IL-10 and TNF-$\alpha$ levels were also found to be similar among all groups. Previous studies reported no significant increase of IL-10 and TNF-$\alpha$ in mild cases (*Xu et al., 2020*; *Zhao et al., 2020*), and no differences were reported among patients, recovered individuals, and healthy controls (*Queiroz et al., 2022*; *Phetsouphanh et al., 2022*).

Similar to IL-6, IL-10, and TNF-$\alpha$, CCL-3 and CXCL-13 levels were not significantly different between mild-to-moderate COVID-19 patients, recovered individuals, and

healthy controls. Previous studies revealed no significant difference in CCL-3 levels between mild and severe infection, and between non-hospitalized and non-ICU hospitalized patients (*Xu et al., 2020*; *Noto et al., 2022*). Six months post-infection, the levels of both CCL-3 and CXCL-13 were not elevated in either hospitalized or non-hospitalized individuals (*Noto et al., 2022*).

The levels of IL-6, IL-10, TNF- $\alpha$, CCL-3, and CXCL-13 might not be a concern among mild-to-moderate SARS-CoV-2 infected and recovered pre-vaccinated Indonesian individuals. However, elevated IL-8, CCL-2, and CCL-4 might need to be considered. Chronic elevation of pro-inflammatory cytokines has been hypothesized to be involved in the pathophysiology of post-acute sequelae of COVID-19 (PASC) or Long COVID, including persistent dyspnea due to lung damage and impaired lung function, fatigue, malaise, and autonomic dysfunction (*Klein et al., 2022*; *Gavriilaki & Kokoris, 2022*).

As for the sex, the differences in IL-10 levels between SARS-CoV-2 infected individuals and those healthy or recovered are not fully understood. However, research has shown that SARS-CoV-2 infection can lead to sex-related differences in immune responses. A study found that male patients had higher plasma levels of innate immune cytokines such as IL-8 and IL-18, while female patients had a more robust T cell activation during SARS-CoV-2 infection (*Takahashi et al., 2020*). These findings suggest that there may be a more robust ability among women to control infectious agents, but the specific reasons for the gender differences in IL-10 levels remain to be determined. Further research is needed to fully understand the underlying mechanisms for these gender differences in immune responses to SARS-CoV-2 infection. Although both innate and adaptive immunity are important in the fight against foreign antigens, study suggests that the adaptive immunity in female patients is more robust and sustained in females compared to male patients. IL-10 is well known for its beneficial roles in inflammation resolution and tissue repair in inflammatory diseases. The higher level of IL-10 in female *vs.* male patients may be an important reason why female patients have a better prognosis and lower mortality following SARS-CoV-2 infection (*Qi et al., 2021*).

The CXCL-13 levels were negatively correlated with age. Previous study assigned CXCL-13 as a signature chemokine for long COVID due to its lower levels among individuals with long COVID compared to those who recovered without exhibiting persistent symptoms (*Muri et al., 2023*). CXCL13 production was also correlated to antibody production against multiple SARS-CoV-2 antigens (*Horspool et al., 2021*); therefore, this situation might be correlated with declining acquired immunity against SARS-CoV-2 among recovered older individuals. Previous study revealed that lower antibody titers were correlated to older age (*Dyer et al., 2022*) and lower vaccination response among older Indonesian individuals (*Megasari et al., 2023*). Declining levels of CXCL13 among the elderly are closely related to aging lymph nodes, which show less CXCL13 expression in the follicle. The chemokine plays an important role in the homing of naive B cells into the lymph nodes follicle before undergoing clonal expansion in the germinal center (*Thompson et al., 2019*). Lower CXCL-13 production, especially among older individuals, might need to be considered further due to its possible correlation with lower acquired antibodies and the emergence of symptoms following recovery.

To our knowledge, this study is the first to describe cytokine profiles of SARS-CoV-2 recovered pre-vaccinated Indonesian individuals, especially those of Javanese ethnicity. However, this study has several limitations. First, the sample size of this study is limited. Data acquired in this study was also not normally distributed; therefore, limiting further statistical analysis, such as regression model analysis. Second, this study only involved mild-to-moderate SARS-CoV-2 infection. Cytokine storm and the rise of pro-inflammatory cytokines tend to occur more frequently in severely ill COVID-19 patients (*Tang et al., 2020*; *Kleymenov et al., 2021*). Third, we did not differ the phase of COVID-19 infection. Based on the insights gained from this study, further study in Indonesia with a larger sample size, more diverse ethnicities, and involving COVID-19 patients with various disease severity and recovered individuals with and without PASC is recommended. Fourth, different age distribution was observed among groups, where infected groups and healthy controls were comprised of older participants and younger participants, respectively. Age is known as a potential variable affecting cytokine levels. Older individuals might exhibit different cytokine profiles compared to younger individuals due to inflammaging (*Stowe et al., 2010*; *Michaud et al., 2013*; *Wyczalkowska-Tomasik et al., 2016*).

## CONCLUSIONS

Several cytokine levels, including IL-8, CCL-2, and CCL-4, were significantly elevated in pre-vaccinated Indonesian individuals with mild-to-moderate SARS-CoV-2 infection and those who recovered. Cytokine profiles described in this study might indicate inflammatory responses not only among SARS-CoV-2 infected, but also among recovered individuals. Prolonged inflammation in recovered individuals might mandate further attention due to their probable pathological consequences.

### Funding
This study was funded by the Directorate of Research, Technology and Community Service, Ministry of Education, Culture, Research, and Technology, Indonesia, with grant number 823/UN3.15/PT/2022. The funders had no role in study design, data collection and analysis, decision to publish, or preparation of the manuscript.

### Grant Disclosures
The following grant information was disclosed by the authors:
Directorate of Research, Technology and Community Service, Ministry of Education, Culture, Research, and Technology, Indonesia: 823/UN3.15/PT/2022.

### Competing Interests
The authors declare there are no competing interests.

## Author Contributions

- Ni Luh Ayu Megasari conceived and designed the experiments, performed the experiments, prepared figures and/or tables, authored or reviewed drafts of the article, and approved the final draft.
- Siti Qamariyah Khairunisa performed the experiments, analyzed the data, authored or reviewed drafts of the article, and approved the final draft.
- Radita Yuniar Arizandy performed the experiments, prepared figures and/or tables, and approved the final draft.
- I Komang Evan Wijaksana performed the experiments, authored or reviewed drafts of the article, and approved the final draft.
- Citrawati Dyah Kencono Wungu conceived and designed the experiments, analyzed the data, prepared figures and/or tables, authored or reviewed drafts of the article, and approved the final draft.

## Human Ethics

The following information was supplied relating to ethical approvals (*i.e.*, approving body and any reference numbers):

Ethical Clearance Commission of Health Research, Faculty of Dental Medicine, Universitas Airlangga.

## Data Availability

The raw measurements are available in the Supplementary Files.

## Supplemental Information

Supplemental information for this article can be found online at http://dx.doi.org/10.7717/peerj.17257#supplemental-information.

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
