# Peer review of "Cytokine profiles of mild-to-moderate SARS-CoV-2 infected and recovered pre-vaccinated individuals residing in Indonesia"

_PeerJ, doi:10.7717/peerj.17257_

## Round 0.1 · original submission · Major Revisions

According to the reviewers' comments, your manuscript should be revised and, in some sections, reorganized. Please note that some additions/changes are suggested for ease of reading, accuracy, and/or better logic flow. See more details in each reviewer report.

Reviewer 1 ·

Basic reporting

The manuscript is well written and it is clear. However, the aims of the study need a more powerful justification. The "cytokine storm" occurs in severe COVID-19 cases. Why should the cytokine quantity be important in mild-to-moderate COVID-19 cases? I suggest that the authors highlight the relevance of study performance. In addition, authors should clarify why mild-to-moderate COVID-19 cases were hospitalized or if this was an inaccurate information?

Experimental design

The plasma levels of cytokines could vary according to different factors. Thus, different information must be adequately stated in the manuscript:

- The time when sampling was done. Authors should specify at least the time since symptoms onset or since diagnosis. Especially in moderate cases, the inflammatory process could be expected to be different if patients were on day 1 or day 7 after symptoms onset.

- Likewise, in recovered patients, the authors considered this status when the PCR test was negative, but this is not clear. A patient could have a negative test while still presenting symptoms. If a patient had a negative test after 7 days of symptoms onset, do the authors consider that the inflammatory process is completely finished? Authors should give more information about this. The COVID-19 recovered group is hard to be recognized as it with the information included in the manuscript.

- Other variables that could cause a variation in the inflammatory response could be co-morbidities, body mass index, smoking status, and treatment. I encourage authors to include this information in Table 1 and consider it for the statistical analysis.

- It is important to consider the variability in clinical and demographic characteristics of the three groups of patients. The group of healthy subjects seems to be younger than the COVID-19 group. This should be assessed by a statistical test. Do age, sex or other clinical variables influence cytokine levels?

Other comments:

- Tables should include the n total in the column headings, and the frequency in percentage should be stated n(%) in each cell.

Validity of the findings

I consider that the analysis of the results requires additional efforts. For instance, if the clinical and demographical variables are different among the comparison groups, a multivariable analysis could enrich the results.
In addition, some outliers in the cytokine levels are observed in figures 1 and 2, and this could influence the differences observed. Authors should deep about this in the Discussion Section, to give a more precise interpretation of the results.

Reviewer 2 ·

Basic reporting

In this manuscript, the authors studied cytokine profiles of unvaccinated patients having mild-to-moderate COVID-19 infection and those recovered in Indonesia by comparing them to healthy controls, thus potentially benefiting future studies and patients. Yet, some additions/changes are suggested for ease of reading, accuracy, and/or better logic flow. See more details below.

B1. Based on the Introduction, it seems the authors studied the recovered group in view of the high risk of death over the year following the infection. Yet, the recovered group included in the study seems only to cover those very recently recovered from COVID-19, i.e., still having a positive antibody test result. It would be really appreciated if the authors don’t mind elaborating on this.

B2. It is further suggested to discuss the necessity of analyzing cytokine profiles of unvaccinated populations since vaccine has been used worldwide, including Indonesia.

B3. Concerning the sentence starting in line 52, the virus is designated as SARS-CoV-2 while the disease should be referred to as COVID-19. The language seems to suggest the virus was referred to as COVID-19, too. Kindly make amendments accordingly if appropriate.

B4. Concerning the sentence starting in line 26, it seems the authors meant to mention Indonesia in some part of the sentence.

B5. It is suggested to change “between” in lines 41, 154, and 211 to “among” if appropriate.

B6. It is suggested to use consistent acronyms across the manuscript, such as CCL2 vs CCL-2.

B7. A “not” seems missing in line 196.

Experimental design

E1. Based on the paragraph spanning lines 77-82, it seems the authors meant to focus on “individuals who became symptomatic develop only mild or moderate illness not requiring hospitalization”. Yet, the first tested group described in this manuscript included “24 hospitalized patients”. See, lines 31 and 100. Kindly elaborate on this inconsistency.

E2. It is suggested to include a brief description of how the serum samples were prepared.

E3. Concerning both the infected group and the recovered one, if possible, kindly briefly describe when the serum was collected. Were those from the infected ones obtained upon hospitalization? Have all of the patients in the infected group recovered from COVID-19 eventurally? Were those from the recovered ones obtained upon the first negative NAAT result while confirming their antibody tests remained positive? Were any recovered samples from individuals having both antibody and NAAT tests negative? How often were the patients tested via NAAT? How many asymptomatic individuals were included in the recovered group? What were the main reasons for their testing/hospitalization? What symptoms would make the patient eligible for having mild-to-moderate COVID-19? Also, did any patient contribute to both the infected group and the recovered one? In other words, was there any follow-up result available for a patient who suffered mild-to-moderate COVID-19 and later recovered?

E4. The description spanning lines 121-127 seems unnecessary.

E5. Kindly elaborate on the reasons why only 73 samples were obtained from (24+25+25=) 74 individuals.

E6. It seems the age and gender distributions are different among the three groups. Have the authors detected any age-dependent or gender-dependent cytokine profile differences?

Validity of the findings

F1. The data presented in Figure 1B seems interesting in view of a single outliner in the recovered group. What would the plot and statistical analysis look like if this point is removed? Is there anything special about this subject?

F2. Kindly correct the legend for Figure 2D, while there is no Figure 2E.

·

Basic reporting

In the manuscript submitted by Megasari et al, the authors screened different cytokine levels from three groups including healthy donors, recovered individuals and mild-to-moderate SARS infected patients. They demonstrated that IL-8, CCL2, CCL4 are much higher in infected and recovered group. However, there are many aspects that the author do not provide enough explanation and investigation.
Are there any interesting factor that been identify to differentially expressed between infected and recovered patients?
At which time points were those samples collected? Will it make a difference? Do you keep track of the cytokine expression in the same patient from the beginning of infection?
Do you compare the cytokine levels between female and male? Any difference? And age groups as well
Table 2: missing unit for TNFa
Figure1: please change the display of the figure 1A/B/C to have a better visualization between groups
In general, I think the articles is limited of data and understudied. The authors should investigate and explore more on the data.

Experimental design

N/A

Validity of the findings

N/A

Additional comments

N/A

---

## Round 0.2 · Minor Revisions

Reviewer one has important comments for your revised version.
Please read and amend it carefully; please pay particular attention to the statistical tools and interpretations made.

Reviewer 1 ·

Basic reporting

I appreciate the modifications performed by the authors.

Experimental design

I thank the authors for the additional evaluation of the influence of sex and gender in the cytokines' levels. However, some information is still needed to precise the finding. Which group is included in the analysis presented in Table 3? Is this was only performed for one group, or the three groups were mixed? It is important to consider that there were differences in the age between groups. Please, add this explanation.
In addition, in the Results Section the results of the Table 3 should be detailed. Authors state: "...while CCL-4 levels were different between age groups". It is not clear whether the authors performed a correlation test or a comparison of levels between age groups. If a Spearman correlation test was performed, please present the rho values in the text or in the table, at least for the statistical significant result (CCL-4).

Validity of the findings

Authors made additional analysis and found that the CCL-4 levels are influenced by the age (Table 3), and the levels of this cytokine was also different between groups (healthy vs infected, healthy vs recovered). According to figure 2, the cytokine levels are lower in the healthy group when compared to the others two groups, but in this healthy group the youngest subjects can be found. So it is possible that this could be a confounder. Probably a multivariate analysis can help to clarify this difference (regression models), or at least authors should highlight this limitation, and deep in the explanation when the result is presented.

·

Basic reporting

N/A

Experimental design

N/A

Validity of the findings

N/A

Additional comments

N/A

---

## Round 0.3 · Minor Revisions

Please attend to the comments for reviewer 1. Also, review the entire manuscript carefully for grammar and redaction, and be careful with abbreviations and acronyms employed, including tables and figures (if applicable).

Reviewer 1 ·

Basic reporting

I thank the modifications made by the authors.

Experimental design

I appreciate the modifications made by the authors. In the Table 3, I suggest that authors report median and interquartile range since non-parametric test were performed. In addition, if the Spearman's test was used, then the value reported must be a rho or the corresponding symbol instead of the r of Pearson's test.

Validity of the findings

The authors made the corresponding modifications.

---

## Round 0.4 · accepted · Accept

Thank you for addressing all of the reviewers' comments. The current version was improved and is satisfactory for me. This version can be published.